# Metagenomics Insights into the Role of Microbial Communities in Mycotoxin Accumulation During Maize Ripening and Storage

**DOI:** 10.3390/foods14081378

**Published:** 2025-04-16

**Authors:** Xuheng Nie, Xuefeng Chen, Xianli Lu, Shuiyan Yang, Xin Wang, Fuying Liu, Jin Yang, Ying Guo, Huirong Shi, Hui Xu, Xiang Zhang, Maoliang Fang, Yin Tao, Chao Liu

**Affiliations:** 1Yunnan Grain and Oil Science Research Institute, Kunming 650033, China; niekon2014@163.com (X.N.); ysy200891@126.com (S.Y.); ynwx@sina.com (X.W.); 18288226523@163.com (J.Y.); 13888119694@163.com (Y.G.); fml1984@126.com (M.F.); 19988499826@163.com (Y.T.); 2Research Center of Fruit Wine, Qujing Normal University, Qujing 655011, China; 3Sinograin Yunnan Depot Co., Ltd., Kunming 650228, China; 13888973219@163.com; 4Sinograin Qujing Depot Co., Ltd., Qujing 655000, China; 15808743445@163.com (H.S.); 13987400596@163.com (H.X.); 13888580874@163.com (X.Z.)

**Keywords:** maize mycotoxins, microorganisms, metagenomics, producing areas, storage periods

## Abstract

Mycotoxins are among the primary factors compromising food quality and safety. To investigate mycotoxin contamination, microbial diversity, and functional profiles in maize across distinct geographic regions, this study analyzed samples from Xuanwei, Fuyuan, and Zhanyi. Mycotoxin concentrations were quantified through standardized assays, while microbial community structures were characterized using metagenomics sequencing. Metabolic pathways, functional genes, and enzymatic activities were systematically annotated with the KEGG, eggNOG, and CAZy databases. The results demonstrated an absence of detectable aflatoxin (AF) levels. Deoxynivalenol (DON) concentrations varied significantly among experimental cohorts, although all values remained within regulatory thresholds. Zearalenone (ZEN) contamination exceeded permissible limits by 40%. The metagenomic profiling identified 85 phyla, 1219 classes, 277 orders, 590 families, 1171 genera, and 2130 species of microorganisms, including six mycotoxigenic fungal species. The abundance and diversity of microorganisms were similar among different treatment groups. Among 32,333 annotated KEGG pathways, primary metabolic processes predominated (43.99%), while glycoside hydrolases (GH) and glycosyltransferases (GT) constituted 76.67% of the 40,202 carbohydrate-active enzymes. These empirical findings establish a scientific framework for optimizing agronomic practices, harvest scheduling, and post-harvest management in maize cultivation.

## 1. Introduction

Maize is one of the most widely cultivated food crops in the world and plays a crucial role in ensuring global food security. According to the Food and Agriculture Organization of the United Nations (FAO), total global maize production reached 1.2 billion tons in 2023. Additionally, China reports that the country’s maize production was 320 million tons in 2023, making it one of the three major food crops and accounting for approximately 40% of total food production in China. The quality of maize not only affects the survival and health of livestock but also influences the quality of meat products, thereby impacting consumer health through meat, eggs, milk, and other products.

The quality of maize is significantly impacted by mycotoxins, which are among the most harmful factors threatening food quality and safety. Mycotoxins in maize are toxic secondary metabolites known for their strong toxicity and carcinogenic properties, posing serious risks to both human and animal health and leading to food hygiene and safety issues [1]. According to the FAO, approximately 25 percent of the world’s crops harvested each year are contaminated with mycotoxins [2]. In China, maize is also affected by mycotoxin contamination [3]. Mycotoxins can accumulate during various stages of maize production, including growing, harvesting, transportation, and storage; this phenomenon is affected by climate, pest occurrence, farming practices, and other variables [4]. The primary reasons for the accumulation of mycotoxins in maize are as follows. Firstly, changes in planting methods and the prohibition of straw burning to return nutrients to the field have increased the survival rate of soil pathogens, such as Fusarium fungi [5]. Secondly, farmers often fail to dry maize promptly after harvest, and when high moisture levels are present, they tend to use temporary stacking methods for storage. This practice significantly correlates with α and β diversity of fungal communities in maize piles [3]. Furthermore, climate change and extreme weather events have led to a higher incidence of ear rot in maize crops [6]. The contamination of maize with mycotoxins adversely affects its quality during storage and processing, posing serious health risks to both humans and animals and resulting in substantial economic losses. Consequently, the control of fungi and mycotoxins in maize has garnered significant attention from both scientists and consumers.

Researchers have conducted extensive studies on mycotoxin contamination and microbial diversity in maize. These studies have demonstrated that maize is contaminated by many microbial communities and a range of mycotoxins during the growing, harvesting, and storage phases [7]. Fungi that cause significant harm include *Fusarium*, *Aspergillus*, and *Penicillium*. *Fusarium* contamination is prevalent throughout the entire harvest and storage process, making it the most abundant genus found in corn samples. The dominant species within this genus are *F. verticillioides*, *F. graminearum*, and *F. proliferatum* [4]. The quantity of *Fusarium* fungi and the types of mycotoxins they produce vary at different stages of maize processing. Recent studies have indicated that fungal communities exhibit significant differences under various storage conditions [3]. Notably, the diversity of fungi was found to increase significantly after maize was stored in unsealed woven bags compared to sealed ones [7]. Most current studies focus primarily on toxic fungi in maize, neglecting the impact of the microbial community as a whole [8]. The main types of mycotoxins include aflatoxins (AFs), deoxynivalenol (DON), fumonisins (FBs), and zearalenone (ZEN) [9]. The harvested maize was stored in a simulated silo for six months, during which the presence of fungi and their toxins was monitored, revealing an increase in FBs over the storage period [4]. Research conducted both domestically and internationally has indicated that the findings regarding key factors contributing to maize toxin contamination vary across different regions. Mycotoxin contamination can accumulate during the field harvest period, farmer’s storage period, or the storage period in specialized depots [10,11]. There is a scarcity of studies examining the correlation between mycotoxins and microorganisms during maize harvesting and storage. Therefore, systematic and comprehensive research on maize mycotoxin contamination and microbial diversity during corn harvest and temporary storage is essential. Such research can provide valuable data to explore the mechanisms behind maize mycotoxin contamination and facilitate in-depth studies on prevention and control technologies.

Yunnan Province features a complex topography and a diverse climate. The harvest time for maize varies across different regions; the moisture content of maize at harvest is approximately 30%. This high moisture level makes it difficult to dry the maize promptly, creating an environment conducive to the growth of microorganisms and the production of mycotoxins. To obtain a systematic and comprehensive understanding of fungi and mycotoxins during maize harvesting and storage, this study focused on maize from Xuanwei, Fuyuan, and Zhanyi. The mycotoxin content in maize samples was measured, and microbial diversity was analyzed using a metagenomics approach. This research provides a data-driven foundation and theoretical support for maize field foundation, optimal harvest timing, and storage practices.

## 2. Materials and Methods

### 2.1. Materials and Reagents

In this study, the cultivar Dunyu810 of harvested fresh maize was used as the research subject.

AFB_1_, ZEN, and DON standard solutions were purchased from Beijing Meizheng Testing Technology Co., Ltd. (Beijing, China). AFB_1_, ZEN, and DON immunoaffinity columns were obtained from Beijing Hua’an Maike Biotechnology Co., Ltd. (Beijing, China). HPLC-grade methanol and acetonitrile were purchased from Sigma-Aldrich (St. Louis, MO, USA). Purified water was obtained from a water purification system (Sartorius, Germany).

### 2.2. Experimental Design and Sample Collection

Samples were collected from the field at Xuanwei (XWTJ), Fuyuan (FYTJ), and Zhanyi (ZYTJ) in October 2023 and stored in farmers’ houses. From October 17 to 19, 2023, the project team conducted field sampling in Fuyuan County, Zhanyi County, and Xuanwei County. The sampling site in Fuyuan County included four townships and nine villages; the sampling site in Zhanyi County comprised three townships and nine villages; the sampling site in Xuanwei County also involved three townships and nine villages (Figure 1). One maize sample was collected from each village before the harvest in 2023; each sample weighed approximately 4 kg (equivalent to about 10 maize cobs). Three samples were randomly selected from each county for further analysis. Additionally, maize stored by farmers was randomly sampled at 1 day (1DNHCL) and 15 days (15DNHCL) to determine the presence of mycotoxins and microorganisms, with three biological replicates for each treatment.

### 2.3. Extraction and Purification

The extraction and purification of AFB_1_ were conducted in accordance with GB 5009.22-2016 [12]. A 5.00 g crushed sample was weighed, and 25.0 mL methanol-aqueous solution (70:30) was added. The mixture was shaken vigorously for 20 min and then filtered with rapid filter paper. Ten milliliters of the filtrate was removed, and 20.0 mL of deionized water was added for mixing. The mixture was then filtered through glass fiber filter paper, and the filtrate was collected. Prior to purification, the immunoaffinity column was allowed to reach room temperature (15–25 °C) and was then attached to the lower end of a syringe. A precise volume of 15.0 mL of sample filtrate was transferred into the syringe, and the flow rate was adjusted to ensure that the liquid exited at a rate of 1–2 drops per second. Once the liquid was fully drained, the column was washed twice with 10.0 mL of water at a flow rate of 2–3 drops per second until air entered the affinity column. After draining the liquid, 1.0 mL of methanol was added to the column for elution at a flow rate of 1 drop per second, and all eluents were collected. Following this, the eluent was evaporated to near dryness using nitrogen while maintaining a water bath temperature of 40 °C. Subsequently, 1.0 mL of mobile phase was added, and the sample extract was vortexed and filtered using a 0.22 μm filter membrane.

The extraction and purification of DON were conducted in accordance with GB 5009.111-2016 [13]. A 5.00 g crushed sample was weighed, and then 1.00 g of polyethylene glycol and 20.0 mL of pure water were added. The mixture was vigorously shaken for 20 min and filtered using rapid filter paper. The filtrate was then collected using fiberglass filter paper. Prior to purification, the immunoaffinity column was equilibrated to room temperature (15–25 °C) and attached to the lower end of a syringe. A precise volume of 2.0 mL of sample filtrate was transferred into the syringe, and the flow rate was adjusted to allow the liquid to exit at a rate of 1–2 drops per second. Once the liquid was drained, the column was washed twice with 10.0 mL of water at a flow rate of 2–3 drops per second until air entered the affinity column. After draining the liquid, 1.0 mL of methanol was added to the column for elution at a flow rate of 1 drop per second, and all eluents were collected. Following this, the eluent was evaporated to near dryness using nitrogen while maintaining a water bath temperature of 40 °C. Subsequently, 1.0 mL of mobile phase was added, and the sample extract was vortexed and filtered using a 0.22 μm filter membrane.

The extraction and purification of ZEN were conducted in accordance with GB 5009.209-2016 [14]. A 20.00 g sample was weighed, and 100.0 mL of 80% acetonitrile was added. The mixture was shaken vigorously for 20 min and then filtered using rapid qualitative filter paper. A 10.0 mL aliquot of the filtrate was removed and diluted with 40.0 mL of pure water, followed by filtration through glass fiber filter paper to collect the final filtrate. Before purification, the immunoaffinity column was allowed to reach room temperature (15–25 °C) and was connected to a syringe. An accurate volume of 25.0 mL of the filtrate, equivalent to 1.0 g of the sample, was injected into the syringe. The pressure was adjusted to allow the solution to flow at a rate of approximately 1–2 drops per second. After the liquid was drained, the column was washed twice with 10 mL of water at a flow rate of 2–3 drops per second. The column was then eluted with 1.0 mL of methanol at a flow rate of 1 drop per second. The eluent was collected and evaporated to near dryness using nitrogen while maintaining a water bath temperature of 40 °C. Finally, 1.0 mL of the mobile phase was dissolved, and the solution was filtered using a 0.22 μm filter membrane.

### 2.4. Mycotoxins Determination and Apparatus

The AFB_1_ content was determined using HPLC (Agilent 1260) equipped with a FLD and a photochemical post-derivatization system, in accordance with GB 5009.22-2016 [12]. For the quantification of AFB_1_, external standard methods were employed using pure standards at different concentrations. AFB_1_ was separated using an Agilent ZORBAX SB-C18 analytical column (250 mm × 4.6 mm × 5 μm); the mobile phase consisted of 45% methanol and 55% water. The flow rate was set at 1 mL/min, and the injection volume was 20 μL. The detection excitation wavelength was 360 nm, while the emission wavelength was 440 nm.

The content of DON was determined using HPLC with an Agilent 1260 UV detector, in accordance with GB 5009.111-2016 [13]. For the quantification of DON, external standard methods were employed, utilizing pure standards at different concentrations. DON was separated using an Agilent ZORBAX SB-C18 analytical column (250 mm×4.6 mm×5 μm), with the mobile phase consisting of 20% methanol and 80% water. The flow rate was set at 1 mL/min, the injection volume was 20 μL, and the detection wavelength was 218 nm.

The ZEN content was determined using UPLC (Thermo UltiMate 3000) with a FLD in accordance with GB 5009.209-2016 [14]. For the quantification of ZEN, external standard methods were employed, utilizing pure standards at different concentrations. ZEN was separated using a Hypersil GOLD analytical column (100 mm × 2.1 mm × 1.9 μm); the mobile phase consisted of 45% acetonitrile, 5% methanol, and 50% water. The flow rate was set at 0.2 mL/min, and the injection volume was 5 μL. The detection excitation wavelength was 274 nm, while the emission wavelength was 440 nm.

### 2.5. Metagenomic Data Analysis

#### 2.5.1. Genomic DNA Extraction

The samples were centrifuged at 7500 r/min for 10 min at 4 °C, and the sediments were ground in liquid nitrogen. DNA extraction was performed using a commercial E.Z.N.A. ®Soil DNA Kit (Omega BioTek, GA, USA) following the manufacturer’s instructions. The integrity and purity of the extracted DNA were assessed using 1% agarose gel electrophoresis and a Nanodrop 2000 (Thermo Fisher Scientific Inc., Waltham, MA, USA) [15].

#### 2.5.2. Library Construction

One microgram of DNA per sample was used as the input material for the DNA sample preparations. Sequencing libraries were generated using the NEBNest^®^ Ultra^TM^ DNA Library Prep Kit for Illumina (NEB, US), following the manufacturer’s recommendations. Index codes were added to attribute sequences to each sample. Briefly, the DNA samples were fragmented, end-polished, A-tailed, and ligated with full-length adapters for Illumina sequencing, followed by further PCR amplification. Finally, the PCR products were purified using the AMPure XP system, and the libraries were analyzed for size distribution with an Agilent 2100 Bioanalyzer and quantified using real-time PCR.

#### 2.5.3. Metagenomic Sequencing

The clustering of the index-coded samples was conducted using a cBot Cluster Generation System, following the manufacturer’s instructions. After cluster generation, the library preparations were sequenced on an Illumina NovaSeq platform, resulting in the generation of paired-end reads.

#### 2.5.4. Genomic Data Preprocessing

The raw read types from metagenomic sequencing were processed using Fastp to obtain clean data for subsequent analysis. If contamination was present in the sample, it was compared with the host database to filter out reads that may have originated from the host, utilizing Bowtie 2 [16]. Assembly analysis was conducted using MEGAHIT v1.2.9 assembly software [17].

#### 2.5.5. Gene Prediction and Abundance Analysis

Based on contigs (≥500 bp) from each sample and the mixed assembly, MetaGeneMark was employed to predict Open Reading Frame (ORF) using default parameters [18]. Predicted genes shorter than 100 nucleotides were excluded from the analysis. To create a non-redundant initial gene catalog, redundancy was eliminated using CD-HIT v4.8.1 software [19]. Clustering was performed with a default identity threshold of 95% and a coverage threshold of 90%, with the longest sequence selected as the representative sequence. Bowtie 2 was utilized to align the clean data from each sample to the initial gene catalog, and the number of reads mapping to each gene in each sample was quantified. Genes with support reads of ≤2 in each sample were filtered out to generate the gene catalog (Unigenes) for subsequent analysis. The abundance of each gene in each sample was calculated based on the number of reads, and the lengths of the genes were compared.

#### 2.5.6. Taxonomic and Gene Functional Annotation

All genes in our catalog were searched in the Non-Redundant Protein Sequence (NR) database with an e-value of ≤8 × 10^−5^ using BLASTP in Diamond v2.1.9.163 software for taxonomic and functional assignment.

To ensure the biological significance of the data, the Lowest Common Ancestor (LCA) algorithm in the MEGAN v6.25.9 software was utilized to obtain the final species annotation information of the sequences. The abundance of a species in a sample was determined by summing the gene abundance associated with that species. Krona analysis, relative abundance overview, abundance clustering thermal map, PCA and NMDS dimensional reduction analysis, Anosim inter-group difference analysis, and LEfSe multivariate statistical analysis of inter-group differential species were conducted based on the abundance tables at each classification level.

For gene function annotation, all unique ORFs were matched with the National Center for Biotechnology Information (NCBI), the Kyoto Encyclopedia of Genes and Genomes (KEGG), the Nonsupervised Orthologous Groups (eggNOG) database, and the Carbohydrate-Active Enzymes (CAZy) database using Diamond [20].

### 2.6. Statistical Analyses

Some of the data in this paper were analyzed by the single-factor three-level design for quantitative data. Post hoc comparisons of significant effects were conducted using the LSD method with a significance level of *p* < 0.05. Before hypothesis testing, a data normality test was performed. If the data were skewed, they were log-transformed. These data analyses were carried out using SAS v. 9.2 (SAS Institute Inc., Cary, NC, USA). The data are presented as the mean ± SE.

## 3. Results

### 3.1. Mycotoxin Content of Maize Samples

In this study, AFB_1_, DON, and ZEN in maize were detected, but AFB_1_ was not found in any of the samples. The DON content varied significantly under different treatments (*F*_4,10_ = 6.45309, *p* = 0.00782). Post hoc analysis revealed that the DON content in the 1DNHCL treatment was significantly higher than that in other treatments (*p* < 0.05, Table 1). The highest content was observed in the DNHCL1 treatment group (662.23 ± 163.19 μg/kg), while the FYTJ treatment group exhibited the lowest (200 ± 0 μg/kg). Importantly, the content of DON did not exceed the safety limit of 1000 μg/kg. The level of ZEN showed no significant differences under various treatment conditions at *p* < 0.05 (*F*_4,10_ = 2.68359, *p* = 0.09364). Post hoc analysis revealed that the ZEN content in the 15DNHCL treatment was significantly higher than that in the XWTJ and FYTJ groups, respectively (*p* < 0.05, Table 1). The highest concentration of DNHCL15 was 87 ± 30.89 μg/kg, which surpassed the safety limit of 60 μg/kg, resulting in an over-standard rate of 40.91%. Conversely, the FYTJ treatment group exhibited the lowest concentration at 12.5 ± 1.61 μg/kg (Table 1).

### 3.2. Metagenomic Sequencing Profiles

In this study, MetaGeneMark v3.38 software was utilized to predict the genes of contigs larger than 500 bp. The analysis focused on the number of genes, the total length of genes, the average length of genes, and the proportion of GC bases. The results indicated that there were no significant differences in these metrics among the various treatment groups (Gene Numbers: *F*_4, 10_ = 0.90133, *p* = 0.49874; Total Length: *F*_4, 10_ = 0.88569, *p* = 0.50664; Average Length: *F*_4, 10_ = 4.15114, *p* = 0.03092; GC: *F*_4, 10_ = 0.96058, *p* = 0.46986, Table 2). FYTJ treatment exhibited the highest number of genes (146,409 ± 12,925.63), while the XWTJ treatment displayed the lowest number (130,407.67 ± 12,209.41).

### 3.3. Microbial Community Structures

Five types of microorganisms, i.e., bacteria, fungi, archaea, viruses, and others, were analyzed from the metagenomic data of maize samples. A total of 85 phyla, 129 classes, 277 orders, 590 families, 1171 genera, and 2130 species of microorganisms were detected. Microorganisms with lower relative abundance were grouped into a category (others), and unclassified status (unclassified).

Based on the relative abundance table across various classification levels, the top ten microbial species with the highest relative abundance in each sample (group) were selected. The remaining species were categorized and a relative abundance histogram was created to illustrate the species annotation results for each sample at different classification levels.

The abundance of microorganisms in maize samples was 75.00% at the phylum level. The ten most abundant phyla, listed from largest to smallest, were as follows: *Proteobacteria* (16.99%), *Bacteroidota* (2.85 %), *Actinobacteria* (2.59%), *Firmicutes* (2.13%), *Mucoromycota* (0.17%), *Basidiomycota*(0.12%), *Ascomycota* (0.06%), *Candidatus Bathyarchaeota* (0.04%), *Cyanobacteria* (0.02%), and *Chytridiomycota* (0.02%) (Figure 2A).

The overall abundance of microorganisms in maize samples was 80.88 % at the genus level. The ten most abundant genera, list from highest to lowest were as follows: *Acinetobacter* (7.61%), *Salmonella* (3.69%), *Puteibacter* (2.66%), *Mycobacterium* (1.90%), *Robertmurraya* (1.03%), *Lacticaseibacillus* (0.59%), *Streptomyces* (0.56%), *Aeromonas* (0.40%), *Klebsiella* (0.39%), and *Enterobacter* (0.28%) (Figure 2B).

The abundance of microorganism classified in maize samples was 83.27 % at the species level. the top ten species, ranked from highest to lowest abundance, were *Acinetobacter_baumannii* (4.28%), *Puteibacter_caeruleilacunae* (2.66%), *Salmonella_sp_hn_f5* (2.6%), *Mycobacterium_tuberculosis* (1.90%), *Acinetobacter_bereziniae* (1.25%), *Acinetobacter_sp_YH1901134* (1.24%), *Robertmurraya_kyonggiensis* (1.03%), *Salmonella_sp_zj_f60* (0.74%), *Lacticaseibacillus_paracasei* (0.59%), and *Streptomyces_geysiriensis* (0.42%) (Figure 2C).

### 3.4. Microorganism Community Diversity Among Treatments

The diversity of microbial communities in maize samples was analyzed. The results indicated no significant differences in microbial diversity among the various treatment groups (Chaol: *F*_4,10_ = 0.34686, *p* = 0.84038; ACE: *F*_4,10_ = 0.92344, *p* = 0.48777; Shannon: *F*_4,10_ = 1.43166, *p* = 0.29321; Simpson: *F*_4,10_ = 1.73077, *p* = 0.21941, Table 3).

Through principal coordinate analysis of species composition at the microbial species level in the samples, the variance contribution of PC1 was 76.04%, while that of PC2 was 9.65% (Figure 3). As illustrated in the figure, XWTJ could be distinctly separated from the other groups, which were clustered together. This indicates that the microbial composition in XWTJ differed from that of the other groups at the species level, whereas the microbial compositions among the other groups showed no significant differences.

### 3.5. Function of Microorganism

#### 3.5.1. KEGG Functional Analysis

The genes were annotated using the KEGG database; 73,496 genes were identified. The metabolic pathways were primarily categorized into six groups: biological systems (8347 genes, 11.36%), metabolism (32,333 genes, 43.99%), human diseases (14,435 genes, 19.64%), genetic information processing (6880 genes, 9.36%), environmental information processing (5371 genes, 7.31%), and cellular processes (6130 genes, 8.34%) (Figure 4). Among these pathways, metabolism is the predominant function of maize microorganisms, indicating that the metabolic activity of maize samples is robust, which serves as both a condition and a basis for maize mycotoxin contamination. Within the KEGG secondary metabolic pathways, the global and overview map (12,052 genes, 1.31%), carbohydrate metabolism (5056 genes, 0.55%), and information transduction (4270 genes, 0.47%) represented a significant proportion.

#### 3.5.2. eggNOG Functional Analysis

Among the 370 genes annotated by the eggNOG database, five functional units were identified: A (21 genes, 5.68%), C (42 genes, 11.35%), E (16 genes, 4.32%), L (61 genes, 16.49%), J (12 genes, 4.32%), and O (12 genes, 4.32%). The S functional unit, which had an unknown function, was excluded from this analysis (Figure 5). Notably, the major metabolites, including amino acid transport and metabolism, were consistent with the KEGG annotation results.

#### 3.5.3. CAZy Functional Analysis

There are six main categories in the CAZy database: glycoside hydrolases (GHs), glycosyltransferases (GTs), polysaccharide lyases (PLs), carbohydrate esterases (CEs), auxiliary redox activities (AAs), and carbohydrate-binding modules (CBMs) (Figure 6). A total of 52,437 carbohydrate-active enzymes were identified in maize samples after comparison with the database. Among these six categories, the annotation results for GHs and GTs accounted for 76.67 % (40,202) of the total number of enzymes, indicating that GHs and GTs were the primary enzymes facilitating polysaccharide hydrolysis in carbohydrate compound metabolism.

### 3.6. Microbial Profile Involved in Maize Mycotoxins

In this study, six types of fungi capable of metabolizing food mycotoxins (AFB, DON, ZEN) were identified and selected for further analysis. The results indicated that there was no significant difference in the number of fungi across different treatments (*A*. *terreus*: *F*_4,10_ = 1.08088, *p* = 0.41619; *F*. *graminearum*: *F*_4,10_ = 0.74764, *p* = 0.58145; *F*. *poae*: *F*_4,10_ = 3.19231, *p* = 0.06216; *F*. *culmorum*: *F*_4,10_ = 0.54313, *p* = 0.70812; *F*. *oxysporum*: *F*_4,10_ = 0.86084, *p* = 0.51943; *A*. *alternata*: *F*_4,10_ = 3.07635, *p* = 0.06807, Table 4). Post hoc analysis revealed that the numbers of *F*. *poae* in ZYTJ and FYTJ were significantly higher than that in 15DNHCL. Additionally, the number of *A. alternata* in FYTJ was significantly greater than those in both 1DNHCL and 15DNHCL (*p* < 0.05, Table 4).

## 4. Discussion

### 4.1. Mycotoxins in Maize and Their Influencing Factors

According to the Yunnan Statistical Yearbook, Yunnan’s total grain output in 2023 was 19.74 million tons, of which 10.49 million tons was maize. However, maize is susceptible to infection by toxigenic fungi, leading to mycotoxin contamination that poses a threat to both human and animal health [21]. Therefore, it is crucial to understand the patterns of mycotoxin contamination during the maize harvesting and storage processes employed by farmers. This knowledge is essential for the prevention and control of mycotoxin contamination and ensuring food quality and safety.

In this study, we tested for AFB_1_, DON, and ZEN in harvested fresh maize, finding DON and ZEN. AF is a difuran cyclotoxoid produced by certain strains of *Aspergillus flavus* and *Aspergillus parasiticus*. There are approximately 20 derivatives of AF, including B_1_, B_2_, G_1_, G_2_, M_1_, M_2_, GM, P_1_, Q_1_, and others. The four major naturally occurring AFs are AFB_1_, AFB_2_, AFG_1_, and AFG_2_. The order of toxicity for these, from acute to chronic, is as follows: B_1_>G_1_>B_2_>G_2_ [22]. Although tested for AFB_1_, it was not found in our samples. The absence of AFB_1_ in the field samples may be attributed to two factors: it may not have reached the necessary production conditions, or there may be an antagonistic interaction with the metabolism of other mycotoxins, which had an inhibitory effect on the metabolism of AF. Our results are inconsistent with those of other studies; for instance, DON and ZEN were not detected in maize harvested in Spain from 2016 to 2018 [23], nor were they found in studies conducted in Portugal [4]. In contrast, DON was identified in maize samples from India [24], Brazil [25], and Serbia [26]. We detected both ZEN and DON in our maize samples, which aligns with previous research indicating that ZEN and DON often appear simultaneously in grains due to the presence of identical metabolic fungi [27]. These findings highlight the variability in maize microbial communities and mycotoxin contamination across different regions. These discrepancies in results may be attributed to factors such as climatic conditions, genetic backgrounds, and other influences [28]. 

When maize was stored by farmers for 15 days after harvest, the level of ZEN increased significantly. This finding aligns with the results of a previous study [29]. In most cereal crops, AF contamination occurs post-harvest, and levels can escalate rapidly during storage and transportation due to improper management, such as high temperatures and relative humidity (>65%) [29]. While ZEN production primarily occurs in the field, it can also be synthesized under poor grain storage conditions, particularly when moisture levels exceed 30% to 40% [30]. Therefore, to ensure that freshly harvested grain can be stored for an extended period without contamination by mycotoxins, it is essential to control the storage temperature, humidity, and moisture content. 

AFB_1_ is classified as a group 1 carcinogen to humans by the International Agency for Research on Cancer (IARC) [31]. It can induce tumors and other negative pathological conditions, as well as suppressing immune function [32]. Consumption of AF-contaminated food can lead to various diseases, with severity depending on the amount and duration of exposure. These diseases are generally referred to as aflatoxicosis, and acute symptoms in humans include vomiting, hemorrhage, abdominal pain, jaundice, pulmonary and cerebral edema, coma, convulsions, and even death [33]. Consequently, the contamination of AFs in human food is strictly regulated by the U.S. Food and Drug Administration (FDA), which mandates that AF levels in cereals and cereal-based foods should be less than 20 ug/kg. With regard to the acute effects of toxicity, potential exposure to foods contaminated with DON can induce symptoms such as nausea, vomiting, diarrhea, headache, dizziness, abdominal pain, and fever [34]. In addition to its acute toxicity, DON has been shown to disrupt gut morphology and function in animal studies following prolonged exposure to low doses of DON-contaminated food [35]. The strong estrogenic activity of ZEN is a significant public health concern, as it can bind to estrogen receptors, leading to reproductive issues in certain animals and potentially in humans [36,37]. Sporadic epidemics have indicated that ZEN may be a contributing factor to the development of central precocious puberty (CPP) and could cause premature puberty in children [38]. Similar to DON, ZEN is classified as a group 3 carcinogen by the IARC [31]. Therefore, the prevention and control of mycotoxins in grain can effectively mitigate the threat of these toxins to human health.

### 4.2. Composition and Function of Microorganisms

To prevent and control mycotoxins in grain while ensuring quality and safety, it is essential to inhibit the growth of toxigenic fungi. Some studies on fungal growth in grain piles have only assessed the composition of fungi using traditional methods. However, these approaches have limitations that hinder a comprehensive understanding of the microbial composition in grain piles [3,39]. To further analyze the microbial composition of toxic fungi in maize in the field, we employed metagenomics to identity microorganisms in maize samples. Compared to traditional methods, metagenomics overcomes the technical limitations of microbial isolation and culture, allowing for the direct extraction of DNA from samples for sequencing. This approach provides more comprehensive information, including classification, community structure, gene function, and metabolic networks of microorganisms within the samples. Such insights are crucial for understanding the mechanisms behind mycotoxins in maize sample [40,41,42]. The results of α diversity indicated that there were no significant differences in the Chao index, Shannon index, and Simpson index among the various treatments. However, the ACE index after 15 days of storage was greater than after 1 day of storage. These findings suggest that there was no significant difference in microbial diversity among the samples. Alpha diversity refers to the diversity within a specific region or ecosystem and serves as a comprehensive indicator of both richness and evenness. It is primarily influenced by two factors: the number of species (richness) and the uniformity of the distribution of individuals within a community (diversity). The community richness indices mainly include the Chao index and ACE index, while the community diversity indices primarily consist of the Shannon index and Simpson index [43].

Function annotation indicates that carbohydrates are the primary components of cell structure and energy supply substances. They regulate cellular activities and provide the material basis for other metabolic processes. Consequently, carbohydrate metabolism constitutes the second largest category within the metabolic classification (Figure 4). The CAZy database encompasses a diverse array of enzymes that play active roles in the synthesis and degradation of polysaccharides and glycan conjugates. The enzymes available in this database include GH, GT, PL, CE, CBM, and AA [29]. GH comprises a large group of enzymes involved in the metabolism of polysaccharides such as starch, cellulose, lignin, and chitin and is responsible for the hydrolysis of glycosidic bonds that link two or more carbohydrate and non-carbohydrate components [44,45]. GT can catalyze the formation of active oligosaccharides or glycosidic bonds to create various receptors, including proteins, nucleic acids, oligosaccharides, lipids, and small molecule combinations [46]. Although CBM itself does not exhibit enzymatic activity, it can enhance the activities of all other CAZy enzymes [47]. As shown in Figure 4, metabolic genes account for 43.99% of the total. The results presented in Figure 5 indicate that the microorganisms in the sample demonstrated strong metabolic activity and vitality. Furthermore, Figure 6 illustrates that glycoside hydrolases and glycosyltransferases play a dominant role in this process. This dominance may be linked to the decomposition of polysaccharides in maize samples by microorganisms for their own reproduction; however, further studies are required to confirm this hypothesis.

### 4.3. The Toxigenic Fungi in Maize Samples

Mycotoxins are secondary metabolites produced by toxigenic fungi under suitable conditions. In our study, toxigenic fungi were detected in maize samples. Aspergillus species are the primary producers of AFs through a polyketide pathway [48]. Previous studies have identified the common aflatoxin-producing fungi in grains as *A. flavus*, *A. parasiticus*, *A. terreus*, and *A. nomius* [49,50]. Fungi that produce ochratioxin A include *A. ostianus*, *A. ochraceus*, *A. niger*, *A. awamori*, *A. fumigatus*, *A. versicolor*, *A. sclerotiorum*, *A. glaucus*, *A. tubingensis*, *A. westerdijkiae*, *P. verrucosum*, *P. oxalicum*, and *P. brevicompactum* [50,51]. DON is a naturally occurring mycotoxin primarily produced by Fusarium graminearum and Fusarium culmorum in the field and/or during postharvest storage [52]. Related microorganisms include *F. graminearum*, *F. asiaticum*, *F. equiseti*, *F. incarnatum*, *F. poae*, *and F. culmorum* [53,54]. The fungus that produces T-2 toxins is *F. sambucinum* [54]. Fumonisin-producing fungi include *F. verticillioides*, *F. proliferatum*, *F. fujikuroi*, *F. oxysporum*, *A. niger*, and *A. westerdijkiae* [49,54]. ZEN is predominantly produced by various Fusarium species, such as *F. culmorum*, *F. graminearum*, *F. sporotrichioides* [32], *F. equiseti*, *F. incarnatum*, *F. verticillioides*, *F. sambucinum*, and *F. culmorum* [54,55]. In this study, we detected AF-producing fungi, specifically *A. terreus;* the fungi that produce DON were *F. graminearum*, *F. poae*, *and F. culmorum*. Fumonitin-producing fungi were *F. oxysporum*. The fungi that produce ZEN were *Graminearum*, but there was no significant difference among the different treatment groups (Table 4). From these results, it can be seen that the types of toxic microorganisms in maize may be related to maize varieties and planting areas.

In this study, ZEN increased significantly after 15 days of storage. However, there was no significant difference in the levels of ZEN-producing fungi among the different treatment groups, suggesting that the fungi’s capacity to metabolize the toxin increased during storage. The moisture content of freshly harvested maize is approximately 30%, which occurs during a season of high temperatures. Therefore, to inhibit the growth of microorganisms and the metabolism of harmful microbial substances, it is essential to harvest the maize promptly, reduce its moisture content to below the safe threshold, and store it at a lower temperature.

The results of the present study enhance our understanding of microbial community structure and the accumulation of mycotoxins in maize. Overall, the findings indicate that this maize is associated with a high prevalence of toxic fungi, the accumulation of mycotoxins, and potential hazards to human and animal health. It is of great significance to uncover the relationships among microorganisms, their metabolic pathways, and the associated enzymes, as this will lay a foundation for future studies on the relevant mechanisms of action. Due to the limitations of this study, the sample size was restricted. Therefore, it is recommended that further research be conducted to increase the sample size, allowing for a more comprehensive understanding, the formulation of more objective conclusions, and the identification of optimal harvest times and drying methods to guide agricultural practices.

## 5. Conclusions

This study provides a thorough and comprehensive analysis of the accumulation of microorganisms and mycotoxins during the ripping and storage phases of maize. By employing metagenomic techniques to analyze microbial communities and toxin-producing microorganisms, we identified a total of 2130 microbial species spanning 85 phyla, including six mycotoxigenic fungal species. Additionally, it was observed that maize becomes contaminated with mycotoxins during the field ripping stage. The profiles of these mycotoxins varied; for instance, while DON levels remained within safety limits, ZEN exceeded the standard rate by 40%. Furthermore, we conducted functional analyses of microbial genes using KEGG metabolic pathways, eggNOG gene classification, and CAZy enzyme databases. These analyses and results contribute to a deeper understanding of mycotoxin metabolism mechanisms and hold significant implications for optimizing maize growth conditions, determining appropriate harvest timing, and improving storage management practices.

## Figures and Tables

**Figure 1 foods-14-01378-f001:**
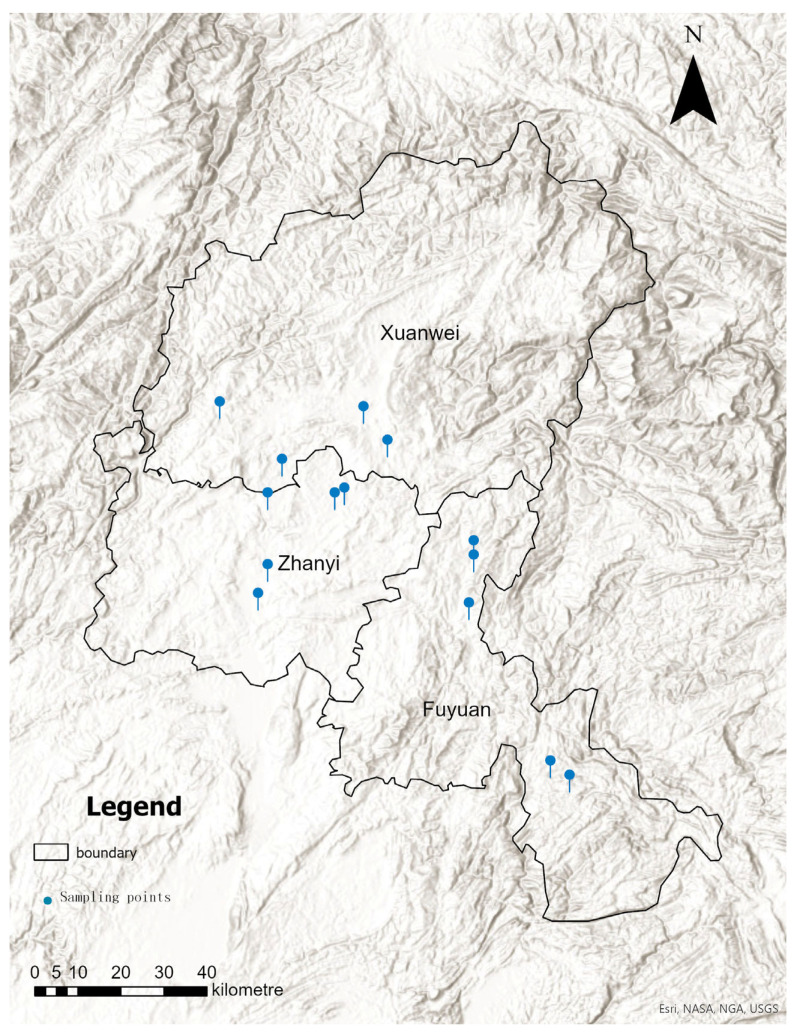
Sampling distribution diagram.

**Figure 2 foods-14-01378-f002:**
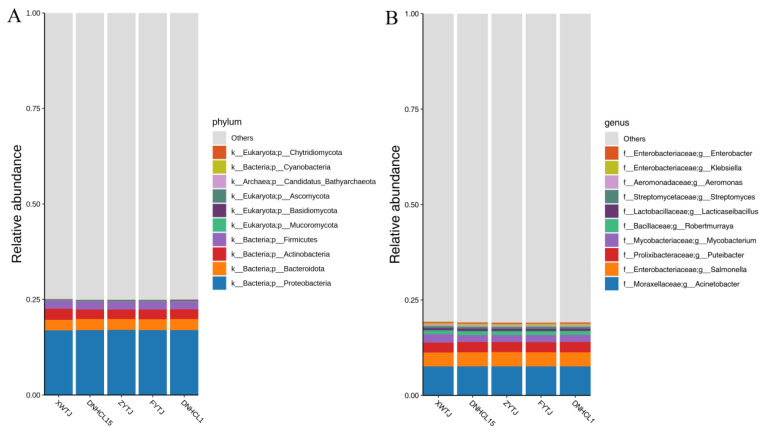
Microbial community structures of maize samples at the phylum (**A**), genus (**B**), and species (**C**) levels.

**Figure 3 foods-14-01378-f003:**
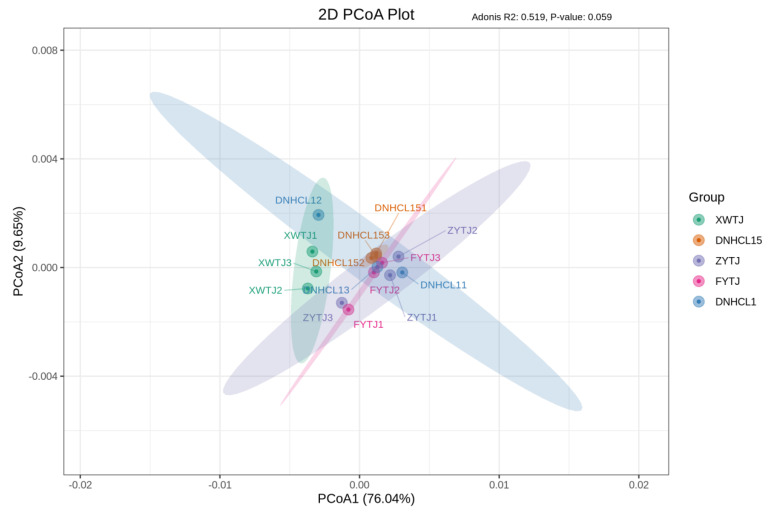
Principal coordinates analysis of microbial community at the microbial species level in maize samples.

**Figure 4 foods-14-01378-f004:**
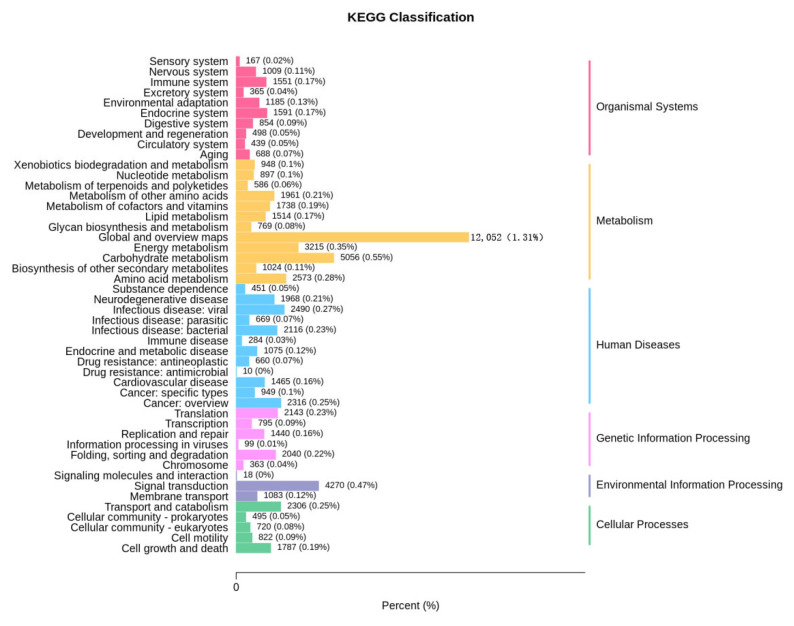
Statistical analysis of microbial gene KEGG metabolic pathway in maize samples.

**Figure 5 foods-14-01378-f005:**
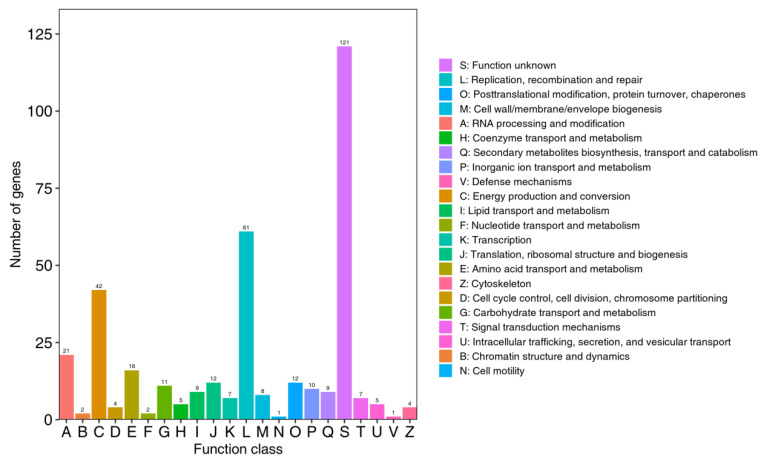
Functional classification of eggNOG gene in maize samples.

**Figure 6 foods-14-01378-f006:**
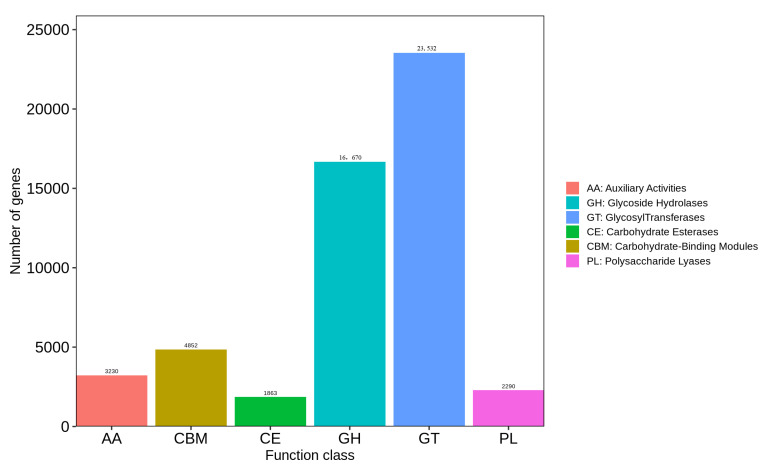
Functional analysis of microbial gene CAZy in maize samples.

**Table 1 foods-14-01378-t001:** Mycotoxin content of maize samples among treatments.

Mycotoxin	Content (μg/kg)				
	XWTJ	ZYTJ	FYTJ	1DNHCL	15DNHCL
AFB_1_	≤1	≤1	≤1	≤1	≤1
DON	247.33 ± 28.03 b	219.27 ± 9.63 b	200 ± 0 b	662.23 ± 163.19 a	321.13 ± 29.32 b
ZEN	21 ± 2.87 bc	47 ± 18.36 ab	12.5 ± 1.61 bc	46.33 ± 16.46 ab	87 ± 30.89 a

Note: The same letters in the table indicate no significant difference among treatment groups.

**Table 2 foods-14-01378-t002:** Non-redundant gene data set.

Samples	Gene Numbers	Total Length	Average Length	GC/%
FYTJ	146,409 ± 12,925.63 a	38.12 ± 3.22 a	260.44 ± 1.06 a	49.35 ± 0.20 a
XWTJ	130,407.67 ± 12,209.41 a	33.93 ± 3.05 a	260.21 ± 1.07 a	48.28 ± 0.29 a
ZYTJ	137,473.67 ± 13,373.03 a	36.04 ± 2.99 a	262.44 ± 3.84 a	49.48 ± 0.33 a
1DNHCL	132,814.67 ± 11,180.54 a	35.06 ± 3.08 a	263.91 ± 1.02 a	49.59 ± 0.20 a
15DNHCL	133,184.67 ± 6713.58 a	35.42 ± 1.63 a	265.97 ± 1.80 a	49.58 ± 0.15 a

Note: The same letters in the table indicate no significant difference among treatment groups.

**Table 3 foods-14-01378-t003:** Species level microbial community α diversity index.

Samples	Chaol Indices	ACE Indices	Shannon Indices	Simpson Indices
FYTJ	1415.39 ± 68.77 a	1381.67 ± 28.36 a	1.124 ± 0.003 a	0.369 ± 0.001 a
XWTJ	1393.32 ± 54.51 a	1328.23 ± 12.59 a	1.128 ± 0.001 a	0.371 ± 0.001 a
ZYTJ	1368.82 ± 181.08 a	1313.70 ± 100.81 a	1.122 ± 0.004 a	0.369 ± 0.002 a
1DNHCL	1401.87 ± 87.90 a	1387.38 ± 91.42 a	1.124 ± 0.003 a	0.370 ± 0.002 a
15DNHCL	1401.20 ± 71.02 a	1401.20 ± 71.02 a	1.124 ± 0.003 a	0.369 ± 0.001 a

Note: The same letters in the table indicate no significant difference in microbial community diversity index among treatment groups.

**Table 4 foods-14-01378-t004:** The profiles of toxic fungi in maize samples.

Species	Contents				
	XWTJ	ZYTJ	FYTJ	1DNHCL	15DNHCL
*A. terreus*	5.33 ± 1.86 a	11.67 ± 3.51 a	7.33 ± 4.06 a	10.67 ± 1.76 a	9.33 ± 1.76 a
*F. graminearum*	10.33 ± 3.84 a	21.33 ± 4.81 a	46.00 ± 35.59 a	25.33 ± 10.74 a	10.33 ± 3.67 a
*F. poae*	4 ± 0 ab	7.33 ± 1.76 a	6.67 ± 0.67 a	4.67 ± 0.67 ab	2.67 ± 1.33 cb
*F. culmorum*	10.67 ± 8.74 a	11.33 ± 8.51 a	24.67 ± 17.29 a	5.33 ± 3.53 a	8.33 ± 6.89 a
*F. oxysporum*	10 ± 3.46 a	15.00 ± 2.51 a	9.00 ± 2.52 a	15.33 ± 5.46 a	15.33 ± 1.86 a
*A. alternata*	12.00 ± 2.65 ab	12.67 ± 5.04 ab	17.33 ± 3.28 a	5.67 ± 1.20 bc	4.67 ± 0.67 bc

Note: The same letters in the table indicate no significant difference in the number of fungi among treatment groups.

## Data Availability

The original contributions presented in the study are included in the article; further inquiries can be directed to the corresponding author.

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
