# Peer review of "Metagenomics Insights into the Role of Microbial Communities in Mycotoxin Accumulation During Maize Ripening and Storage"

_foods, 2025, doi:10.3390/foods14081378_

Round 1
Reviewer 1 Report
Comments and Suggestions for Authors
Interesting paper. We need to know specific conditions for the fields and harvesting periods. Why in 2023 only? What was the repetition?
In this study, the harvested fresh maize in the field was used as the research object. 96 Samples were collected from the field at Xuanwei (XWTJ), Fuyuan (FYTJ) and Zhanyi 97 (ZYTJ) in October 2023, and stored in farmers for 1 day.
What do you mean stored in farmers?
One sample of maize from the field before harvest in 2023 was collected from each village, a total of 27 samples were collected after harvest you mean?
Need to have references for extraction, purification.
Methods need to be written in passive voice.
Methods/methodologies do not have references cited.
What maize cultivar have the authors used?
Please use a map of the different areas.
Metagenomic analysis is fine.
Comments on the Quality of English LanguageNeeds to be checked by a native English person.
Author Response
Apr. 4, 2025
Dear Editor,
Thank you very much for your letter and the comments about our manuscript, Metagenomics Insights into the Role of Microbial Communities in Mycotoxin Accumulation during Maize Ripening and Storage (foods-3521408), submitted to Foods. We also wish to take this opportunity to thank the reviewers for their constructive comments and valuable recommendations. We have carefully revised the manuscript according to the comments and an item by item response is as follows (our answers are in blue for easy identification; modifications based on the comments are in red in revised manuscript).
Thanks for all the help.
Best wishes,
Chao Liu
Qujing Normal University, Qujing, P.R. China
E-mail address: Liuchao_80@163.com.
Reviewer #1:
We would like to thank Reviewer #1 for these constructive comments and valuable recommendations which are of great help to our revision work. We have checked the whole manuscript. Below is our point-by-point response to the reviewer’s comments, we are trying our best to revise the manuscript and hope that the correction will meet with approval.
Comments:
- Interesting paper. We need to know specific conditions for the fields and harvesting periods. Why in 2023 only? What was the repetition?
Answer: Thanks for the reviewer’s comments. The harvest time of the corn samples was October 2023. The detailed description is provided in Section 2.1 on page 3 of the article. One of the shortcomings of this study is that the reproducibility in terms of sample quantity and time is insufficient. Due to the lack of manpower and resources, we only conducted biological replicates in different regions within the same year to make the results more in line with the objective reality. For a detailed description, please refer to Section 2.2.
- In this study, the harvested fresh maize in the field was used as the research object. 96 Samples were collected from the field at Xuanwei (XWTJ), Fuyuan (FYTJ) and Zhanyi 97 (ZYTJ) in October 2023, and stored in farmers for 1 day.
Answer: Thanks for the reviewer’s suggestion. We have revised our manuscript according to your recommendations. For a detailed description, please refer to Section 2.1 and 2.2. Thank for the reviewer’s comment again.
- What do you mean stored in farmers?
Answer: Thanks for the reviewer’s suggestion. We have revised our manuscript according to your recommendations. For a detailed description, please refer to Section 2.1 and 2.2. Thank for the reviewer’s comment again.
- One sample of maize from the field before harvest in 2023 was collected from each village, a total of 27 samples were collected after harvest you mean?
Answer: Thanks for the reviewer’s suggestion. We have revised our manuscript according to your recommendations. For a detailed description, please refer to Section 2.1 and 2.2. Thank for the reviewer’s comment again.
- Need to have references for extraction, purification.
Answer: Thanks for the reviewer’s suggestion. We have revised our manuscript according to your recommendations. For a detailed description, please refer to Section 2.3. Thank for the reviewer’s comment again.
- Methods need to be written in passive voice.
Answer: Thanks for the reviewer’s suggestion. We have revised our manuscript according to your recommendations. Thank for the reviewer’s comment again.
- Methods/methodologies do not have references cited.
Answer: Thanks for the reviewer’s suggestion. We have revised our manuscript according to your recommendations. For a detailed description, please refer to Section 2.3 and 2.5. Thank for the reviewer’s comment again.
- What maize cultivar have the authors used?
Answer: Thanks for the reviewer’s suggestion. We have revised our manuscript according to your recommendations. Thank for the reviewer’s comment again.
- Please use a map of the different areas.
Answer: Thanks for the reviewer’s suggestion. We have revised our manuscript according to your recommendations. For a detailed description, please refer to Section 2.1. Thank for the reviewer’s comment again.
- Metagenomic analysis is fine.
Answer: Thanks for the reviewers' affirmation of the research work.
- Comments on the Quality of English Language
Needs to be checked by a native English person.
Answer: Thanks for the reviewer’s suggestion. We have revised our manuscript according to your recommendations. Thank for the reviewer’s comment again.

Reviewer 2 Report
Comments and Suggestions for Authors
The manuscript titled “Metagenomics Insights into the Role of Microbial Communities in Mycotoxin Accumulation During Maize Ripening and Storage” approached to identify responsible microbes to mycotoxin accumulation in maize at Ripening and Storage conditions… though the work is sound but there are several issue should be revised as follows:
”Abstract: please concisely narrate which microbes were abundant in the different samples of maize. Of course it should be revised /edited according to the results.
Methods: AFB1, ZEN and DON standard solution?? This not clear as its mentioned as colum, standard solution, and in 2.3. Extraction and Purification its seems plant samples??
This is a great confusion to me, therefore it should clarified to all readers
2.4. Mycotoxins Determination and Apparatus:
The AFB1, DON, Zen content were determined using HPLC, UPLC: interpreatation is not clear…therefore writing is so much confusing at the methodology section…either it should be simplified and unique for all contents of AFB1, DON, Zen.
Method sections did not use any references which is great lacking of the manuscript
The conclusion section should not repeat the sentences written in abstract
The language is sometimes very much confusing to understand the storey, therefore language proofreading should be donbe.
Comments on the Quality of English Languageprofessional proofread should be done
Author Response
Apr. 4, 2024
Dear Editor,
Thank you very much for your letter and the comments about our manuscript, Metagenomics Insights into the Role of Microbial Communities in Mycotoxin Accumulation during Maize Ripening and Storage (foods-3521408), submitted to Foods. We also wish to take this opportunity to thank the reviewers for their constructive comments and valuable recommendations. We have carefully revised the manuscript according to the comments and an item by item response is as follows (our answers are in blue for easy identification; modifications based on the comments are in red in revised manuscript).
Thanks for all the help.
Best wishes,
Chao Liu
Qujing Normal University, Qujing, P.R. China
E-mail address: Liuchao_80@163.com.
Reviewer #2:
We would like to thank Reviewer #2 for these constructive comments and valuable recommendations which are of great help to our revision work. We have checked the whole manuscript. Below is our point-by-point response to the reviewer’s comments, we are trying our best to revise the manuscript and hope that the correction will meet with approval.
Comments:
The manuscript titled “Metagenomics Insights into the Role of Microbial Communities in Mycotoxin Accumulation During Maize Ripening and Storage” approached to identify responsible microbes to mycotoxin accumulation in maize at Ripening and Storage conditions… though the work is sound but there are several issue should be revised as follows:
- Abstract: please concisely narrate which microbes were abundant in the different samples of maize. Of course it should be revised /edited according to the results.
Answer: Thanks for the reviewer’s suggestion. Our research did not reveal any significant differences in microbial composition among different corn samples. Therefore, we did not elaborate much on this aspect.. Thank for the reviewer’s comment again.
- Methods: AFB1, ZEN and DON standard solution?? This not clear as its mentioned as colum, standard solution, and in 2.3. Extraction and Purification its seems plant samples??
This is a great confusion to me, therefore it should clarified to all readers
Answer: Thanks for the reviewer’s suggestion. We have revised our manuscript according to your recommendations. For a detailed description, please refer to Section 2.3. Thank for the reviewer’s comment again.
- 2.4. Mycotoxins Determination and Apparatus:
The AFB1, DON, Zen content were determined using HPLC, UPLC: interpreatation is not clear…therefore writing is so much confusing at the methodology section…either it should be simplified and unique for all contents of AFB1, DON, Zen.
Answer: Thanks for the reviewer’s comment. Since our research institute has different liquid chromatographs and these mycotoxins are detected on different instruments, different types of instruments are mentioned in the text. Thank for the reviewer’s comment again.
- Method sections did not use any references which is great lacking of the manuscript
Answer: Thanks for the reviewer’s suggestion. We have revised our manuscript according to your recommendations. For a detailed description, please refer to Section 2.3 and 2.5. Thank for the reviewer’s comment again.
- The conclusion section should not repeat the sentences written in abstract
Answer: Thanks for the reviewer’s suggestion. We have revised our manuscript according to your recommendations. For a detailed description, please refer to Section 5. Thank for the reviewer’s comment again.
- The language is sometimes very much confusing to understand the storey, therefore language proofreading should be donbe.
Answer: Thanks for the reviewer’s suggestion. We have revised our manuscript according to your recommendations. Thank for the reviewer’s comment again.
- Comments on the Quality of English Language
professional proofread should be done
Answer: Thanks for the reviewer’s suggestion. We have revised our manuscript according to your recommendations. Thank for the reviewer’s comment again.

Round 2
Reviewer 1 Report
Comments and Suggestions for Authors
authors have revised sufficiently and paper can be accepted
Comments on the Quality of English LanguageNeeds another check
Reviewer 2 Report
Comments and Suggestions for Authors
well responded to the comments, therefore can be accepted for publication